# Shifting from Conventional to Organic Filter Media in Wastewater Biofiltration Treatment: A Review

Zhang Zhan Loh [1], Nur Syamimi Zaidi [1,2], Achmad Syafiuddin [3,*], Ee Ling Yong [1], Raj Boopathy [4,*], Ahmad Beng Hong Kueh [5] and Dedy Dwi Prastyo [6]

1   School of Civil Engineering, Faculty of Engineering, Universiti Teknologi Malaysia (UTM),
    Johor Bahru 81310, Johor, Malaysia; zhangzhanloh@gmail.com (Z.Z.L.); nursyamimi@utm.my (N.S.Z.);
    eeling@utm.my (E.L.Y.)
2   Centre for Environmental Sustainability and Water Security (IPASA), Universiti Teknologi Malaysia,
    Johor Bahru 81310, Johor, Malaysia
3   Department of Public Health, Universitas Nahdlatul Ulama Surabaya, Surabaya 60237, Indonesia
4   Department of Biological Sciences, Nicholls State University, Thibodaux, LA 70310, USA
5   Department of Civil Engineering, Faculty of Engineering, Universiti Malaysia Sarawak,
    Kota Samarahan 94300, Sarawak, Malaysia; kbhahmad@unimas.my
6   Department of Statistics, Institut Teknologi Sepuluh Nopember, Surabaya 60111, Indonesia;
    dedy-dp@statistika.its.ac.id
*   Correspondence: achmadsyafiuddin@unusa.ac.id (A.S.) or ramaraj.boopathy@nicholls.edu (R.B.)

**Abstract:** Biofiltration is a promising wastewater treatment green technology employed to remove various types of pollutants. The efficiency of biofiltration relies on biofilm, and its performance is significantly influenced by various factors such as dissolved oxygen concentration, organic loading rate, hydraulic retention time, temperature, and filter media selection. The existing biofilters utilize conventional media such as gravel, sand, anthracite, and many other composite materials. The material cost of these conventional filter materials is usually higher compared to using organic waste materials as the filter media. However, the utilization of organic materials as biofilter media has not been fully explored and their potential in terms of physicochemical properties to promote biofilm growth is lacking in the literature. Therefore, this review critically discusses the potential of shifting conventional filter media to that of organic in biofiltration wastewater treatment, focusing on filtration efficiency-influenced factors, their comparative filtration performance, advantages, and disadvantages, as well as challenges and prospective areas of organic biofilter development.

**Keywords:** biofiltration; biofilm; organic filter media; wastewater treatment; green technology

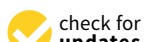



## 1. Introduction

Lately, the presence of excess nitrogen in wastewater receives attention as it promotes the advancement of eutrophication and other hazardous effects on bodies of water [1]. Likewise, the continually diminishing freshwater resources worldwide coupled with depleting accessibility to clean water has inflicted water scarcity issues in both developed and developing countries [2,3]. To prevent further supply deterioration of this necessity, various techniques to remedy wastewater have been introduced either via physicochemical or biological treatments. Conventional physicochemical treatments include aeration, chemical oxidation, coagulation-flocculation, filtration, and ion exchange while biological methods are comprised of activated sludge with various modifications [4–6], aerobic digestion, lagoons, and biofiltration. Despite their wide variety, these physical-chemical treatments have many drawbacks, such as chemical consumption, higher sludge production, higher investment, and capital cost, as well as a high energy requirement [7]. Therefore, studies that investigate the use of biological processes in water and wastewater treatment are conducted due to their advantages such as energy-saving, operation flexibility, lower cost, environmentally friendly, and higher retention capacity [7–17].

Amongst the existing biological treatments, the process using biofiltration has been broadly favored by various researchers in the past few years due to its promising potential. This system is normally used as a tertiary treatment process in the removal of nutrients, toxins, and recalcitrant compounds [18]. Biofiltration, which is an attached growth system, is different from the suspended treatment process as there is a clear separation between the treated effluent and the microbial biomass present in the biofilm. The microbial biomass present in the biofilter is immobilized to the filter bed while the effluent flows through the filter media, the event of which creates a separation between the microbial biomass and the effluent [19]. Biofilters are widely applied in the treatment of nitrogenous and organic pollutants in municipal wastewater treatment plants due to their great efficiency in handling various kinds of water, for instance, oil and gas produced water [20], river water [21], raw sewage [22], groundwater [23], and domestic wastewater [24,25]. Studies have also been focused on operating biofilters under different conditions such as filter media, temperatures, backwash regimes, and dissolved oxygen concentrations to achieve remarkable pollutant removal performance [26–28]. Even so, the cleaning mechanism of the biofilter is somewhat complex, thereby requiring further investigation before an optimally advanced biofiltration system can be developed.

Other than that, there exists an increasing interest in exploring suitable alternative filter materials such as wood chips, wheat straws, and natural fibers [29–31] to replace conventional filter media such as sand, gravel, and anthracite in order to reduce the operational and investment costs of the biofiltration treatment system. This is further motivated by the local availability of these proposed organic materials and their similar physicochemical properties compared to the conventional synthetic media such as polystyrene.

Although there are review papers [32–35] discussing some aspects of biofilters, a comprehensive review on the challenges and prospects from the perspective of shifting the use of conventional filter media to that of organic is missing. Therefore, this review paper fills the intellectual gap by addressing the factors affecting the biofiltration process, the potential use of organic material such as from those of agro-waste as the filtration media, and the prospects of wastewater treatment by biofilters.

## 2. Factors Affecting the Efficiency of Biofiltration

Biofiltration is a biological treatment process that involves a series of complex cleaning mechanisms and operating conditions. The performance of biofiltration is largely influenced by various factors such as dissolved oxygen concentration, organic loading rate, hydraulic retention time, temperature, and filter media (Figure 1). Therefore, it is important to capture the influence of these factors in the development of efficient biofilters.

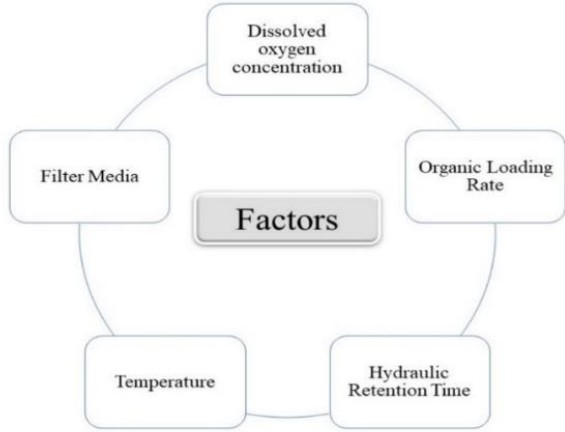

**Figure 1.** Factors affecting the efficiency of a biofiltration treatment system.

Maintaining enough dissolved oxygen concentration is important in the establishment of biological activity to promote biodegradation in a biofilter. The amount of dissolved

oxygen largely affects the growth of the microbial community in the biofilm and the overall efficiency of the biological process. For example, aerobic microorganisms need sufficient dissolved oxygen from an electron acceptor to promote the growth of a biofilm and biodegradation of the pollutant [36]. Hasan et al. [37] observed that adequate oxygen of 2–3 mg/L promotes the growth of nitrifying biofilms and the removal of ammonia in a biological aerated filter. In contrast, a study by Kalkan et al. [38] reported that a low dissolved oxygen concentration (<0.9 mg/L) leads to a lower nitrification efficiency but higher denitrification performance, thus promoting higher total nitrogen (TN) removal. This is because the denitrification process in the biofilter is carried out by heterotrophic denitrifies, where high organic carbon sources and low oxygen concentration favor the growth of this bacteria [39].

Increasing the organic loading rate (OLR) enhances the growth of a biofilm and water holding capacity. This can be explained by the stimulation of microorganisms to promote biological activity under sufficient flux of organic matter to the biofilm [40]. Lee et al. [41] reported the biofilm formation, biomass concentration, and denitrification process all increase with the OLR. However, the nitrification efficiency decreases with an increase in the OLR from 1.0 to 4.0 kg COD/$m^3$ per day. Lefebvre et al. [42] also reported that the treated wastewater effluent meets almost all the discharge regulations under low OLR while increasing the OLR causes foaming issues, which results in an unstable removal performance. As a result of extremely high organic loads that restrict the movement of the substrate into the interior of the biofilm due to the formation of a dense biofilm, the microbial community structure in the biofilm is disrupted [40,43]. Moreover, high dissolved organic matter tends to compete for the adsorption sites with the bacteria in the biofilm, therefore reducing the overall number of available adsorption sites [44].

Hydraulic retention time (HRT) is also one of the important operation factors in preserving the long-term performance of the biofilter. An optimum hydraulic retention time affects the efficiency of the biofilter in terms of cost, as it is directly related to the capacity of the substrate that can be handled per unit time and to the effective contact between the substrate and the microorganism [45,46]. According to Nogueira et al. [47], a fast-growing heterotroph tends to grow in the suspension, so the formation of a biofilm by slow-growing nitrifiers would occur under a comparatively longer HRT. A longer HRT minimizes the competition of dissolved oxygen between the heterotroph and nitrifiers; however, a longer HRT shows disadvantages such as longer treatment time and cost consumption in maintaining the long HRT operational conditions [48].

Besides that, temperature plays a crucial role in controlling and regulating the performance of the biofilter by affecting the growth of the microorganisms in the biofilm. A lower temperature tends to reduce the overall biological process due to lower microbial metabolism and nutrient utilization [49]. Zhang et al. [50] stated that temperature impacts the nitrification and denitrification rate since the growth rate, metabolism, and community structure of nitrifiers, denitrifiers, and the dissolved oxygen level are also affected. A lower temperature lessens the biological treatment efficacy as some bacteria are not suited to survive under low temperatures [51]. It was observed that a low temperature causes a longer acclimation period for the biofilter, such that extra contact time is needed for the less biodegradable compound to meet the specific effluent targets [52]. Zhang et al. [50] reported that the temperature should be set above 18°C to create an ideal environment for nitrifying and denitrifying performance.

Based on the up-to-date review, identification of the optimum factors during the treatment process is crucial in the improvement and development of biofilters, since previous studies are inconsistent in reporting the effects of various factors on the performance of biofilters. This may be attributed to the different types of treated water, operating conditions, and filter media being used in the investigations. Therefore, more studies are recommended to investigate and capture the effects of these factors in achieving the maximum potential of biofilters in the treatment process.

## 3. Filter Media Selection on the Efficiency of Biofiltration

Filter media selection plays an important role in the biofilter as it is one of the main components besides the backwash system and aeration process. Moreover, the oxygen-substrate transfer rate and hydraulic characteristics are affected by the filter media selection [32]. Granular activated carbon (GAC) [53], quartz sand [54], and anthracite [55] are the frequently used conventional filter media in biofilter systems. In recent years, there is a rising interest among researchers in developing and utilizing organic or waste materials as the filter media in the biofiltration treatment process. According to Garzón-Zúñiga et al. [56], several countries, such as the United States and Canada, have applied organic materials in a real-scale decentralized biofiltration treatment system due to several advantages including low operational and construction costs, low maintenance, and does not require a highly-skilled operator to operate the system. Moreover, the utilization of agricultural waste materials as biofilter media will also reduce solid waste residues throughout the world.

### 3.1. Characteristics of Organic Filter Media Compared to Conventional Filter Media

Tejedor et al. [57] stated that organic filter materials should contain favorable characteristics such as larger surface area, higher porosity, physiochemically stable (presence of hemicellulose/lignin content), non-toxic, higher adsorption ability, and contain functional groups (i.e., phenolic hydroxyls, carboxylic or methoxyl). Recently, plant-based wastes are widely utilized by various researchers as support materials for biofilm formation due to advantages such as cost-effectiveness, environmentally friendly, higher specific surface area, higher void fraction, lower bulk density, higher microbial population density, and higher resistance towards biodegradation due to their cellulose, hemicelluloses, and lignin contents [31,58]. Some successful applications of organic materials as biofilm supporting media include Brewer's spent grains, peanut shells, wood chips, wheat straws, rice straws, coconut husk, *Loofah*, *Agave* fibers, and *Arundo donax* [30,31,57,59–62]

Most of the organic biofilm supporting materials are characterized by their cellulose, hemicellulose, and lignin contents. Low et al. [63] identified that high cellulosic and lignin contents increase the strength, durability, and toughness of the plant fibers, which in turn provides the organic media with physicochemical stability, as the structural units of oxyphenylpropanol present in the lignin polymer are relatively hard to be hydrolyzed. Furthermore, these lignocellulose compounds contain polar functional groups such as phenol and carboxyl groups, which tend to increase the adsorption ability, thus promoting the attachment of the biofilm to the organic filter media. The presence of a hydrophilic group on the filter media aids in increasing the water adsorption which enhances the attachment and growth of the biofilm [64]. These chemical properties are similar to those of GAC, which is one of the efficient filter media that contains several functional groups such as carboxylic, alcoholic, and ether groups [65]. Other than that, it was observed that efficient organic filter media commonly show high porosity, i.e., between 74.0 and 84.0% [29,31,57]. The porosity of the organic media is higher compared to that of the conventional media such as anthracite (52.9%) and activated carbon (66.7%) [66]. Higher porosity helps in the prevention of the compaction of the support material in the bioreactor, increases the spaces for fluid circulation, and at the same time, the presence of pores and micropores favorably enhances the conditions of the attachment site for the biofilm [29,31].

Another important parameter of a potential organic media is the surface area of the organic materials. Low et al. [63] demonstrated that the specific surface area of coconut fiber is up to 5.63 m$^2$/g while that of oil palm fibers is up to 2.68 m$^2$/g. These values are higher compared to the specific surface area of anthracite (1.68 m$^2$/g) but lower compared to activated carbon (122.75 m$^2$/g) as reported by Zhang et al. [66]. The superior performance of GAC compared to other media is due to its larger specific surface area in providing more space for the development of a microbial community, which enhances the biofilm formation and improves the biodegradation process [67]. Other than that, the cost of the filter materials is also one of the important parameters influencing the investment and

operational cost of biofilters. Most of the organic filter materials are characterized by their low cost as these organic materials are usually waste or agricultural by-products. It was reported by Saliling et al. [30] that the cost of wood chips and wheat straws are 2.37 US$/m$^3$ and 2.5 US$/m$^3$, respectively, which is much lower compared to the cost of anthracite and GAC at approximately 9.46 US$/m$^3$ and 43.35–47.29 US$/m$^3$ [68], respectively. Vigueras-Cortés et al. [31] reported that the only expenditure for organic *agave* fibers filter media is the transportation cost, as most of the *agave* fibers are available in the form of solid waste.

### 3.2. Conventional vs. Organic Filter Media Biofiltration Systems

Table 1 summarizes the performance of conventional and organic media biofiltration systems. It is well-conceived that biofiltration with conventional filter materials shows remarkable performance in treating various kinds of pollutants such as heavy metals, nitrogenous contaminants, organic compounds, and pharmaceutically active compounds. Conventional biofilter media include sand, anthracite, GAC, zeolite, expanded clay, and plastic media. Among all the conventional filter materials, GAC, sand, and anthracite are the most popular materials. Lately, successful applications of organic waste materials, including peanut shells, coconut fibers, woodchips, rice straws, date palm fibers, and *Agave* fibers as the biofilter media have been reported.

**Table 1.** Performance of conventional and organic media biofiltration system.

| Conventional Media (Natural) | | | | |
|---|---|---|---|---|
| Type of Filter Media | Operating Condition | Removal Efficiency | Remark | Reference |
| Gravel | Temperature: 20 °C<br>pH: 7.0–8.3<br>Airflow rate: 10 L/min<br>DO: 8 mg/L | Mn: 98.0% | Iron should be removed before ammonia and manganese oxidation<br>Start-up: 6 months<br>Treatment of potable water | [69] |
| Sand | Hydraulic flow rate: 0.35–1.56 m$^3$/m$^2$h<br>pH: 6.8–7.2<br>Temperature: 12–15 °C<br>DO: 3 mg/L | COD: 75.0%<br>SS: 97.0%<br>TKN: 62.0% | Natural zeolite shows better nitrogen removal due to the ion exchange capacity with NH$_4$-N. Treatment of textile wastewater | [70] |
| Zeolite (Lab scale) | | COD: 88.0%<br>SS: 97.0%<br>TKN: 80.0% | | |
| Zeolite (Pilot scale) | Hydraulic load: 1.83, 2.3 m$^3$/m$^2$h<br>Temperature: 4–10, 10–18 °C<br>DO: above 2 mg/L | BOD: 99%<br>COD: 92%<br>SS: 74%<br>TN: 92% | | |
| Sand | Filtration rate: 0.015–0.06 m/h<br>Nitrogen loading rate: 8.6–34.3 gN/m$^3$day<br>Surface loading rate: 8.1–32.5 gN/m$^2$day | NO$_3$–N: 94.0% | Start-up period: 1 month<br>Treatment of drinking water | [71] |
| Sand | Iron:arsenic ratio: 10:1, 20:1, 30:1, 40:1<br>Filtration rate: 0.212 m$^3$/m$^2$/h | Removal of arsenic below 5 µg/L | Treatment of drinking water | [72] |
| Anthracite | Filtration velocity: 0.12. 0.25 m/h<br>pH: 7.5 | Turbidity: 50–60%<br>DOC: 21%<br>TN: 50%<br>TP: 36% | Treatment of raw stormwater | [55] |
| Sand | ECBT: 20 min | NH$_4$-N: 46.6–48.5% | Treatment of potable water | [73] |

**Table 1.** *Cont.*

| Conventional Media (Natural) | | | | |
|---|---|---|---|---|
| **Type of Filter Media** | **Operating Condition** | **Removal Efficiency** | **Remark** | **Reference** |
| Quartz Sand | Filtration rate: 3–5 m/h | Iron: 96.2% Manganese: 97.7% Arsenic: 98.2% | Treatment of groundwater | [74] |
| Quartz sand | Filtration rate: 7–9 m/h HRT: 8 min | Ammonia: 73.8% DON: 25.2% | Start-up period: 38 days Treatment of river water (Pre-treated with coagulation and sedimentation) | [66] |
| Anthracite | | Ammonia: 87.9% DON: 3.4% | | |
| Ceramsite | | Ammonia: 76.5% DON: 49.0% | | |
| Combination of anthracite and manganese sand | Temperature: 8 °C pH: 7 | Ammonia: 90.8% Iron: 95.5% Manganese: 95.9% | Start-up period: 81 days Treatment of real groundwater | [75] |
| Sand | HRT: 2 h | Total Organic: 78.0% Ammonium: 82.0% TSS: 91.0% | Start-up period: 1 month Treatment of river water | [54] |
| Sand | Temperature: 20–25 °C Hydraulic load: 1.0 m/d | $NH_4$-N: 60.0% COD: 80.0% | $NO_2$-N accumulation rate: 95.0% Treatment of synthetic wastewater | [76] |
| Quartz Sand | Temperature: 18–23 °C pH: 7.3–7.8 DO: 6–8 mg/L | Fe: near 100% Effluent Mn less than 0.1 mg/L High removal of ammonia | 56 days to achieve the required standard for Fe, Mn, and ammonia Treatment of groundwater | [77] |
| Quartz Sand | HRT: 2 h | $NO_3$-N: 90.0% COD: 75.0% | Start-up period: 20 days Treatment of secondary effluent | [78] |
| Quartz Sand | C/N: 3.7 Temperature: 12 ± 4 °C Filtration rate: 47.1 L/h HRT: 24 min | $NO_3$-N: 74.8% TN: 71.1% $PO_4$-P: 91.2% | Start-up period: 40 days Treatment of synthetic wastewater | [79] |
| Zeolite | Air/water ratio: 1:1 HRT: 3 h Temperature: 10 °C | $NH_4$-N: 95.1% $PO_4$-P: 62.7% | Start-up period: 2 weeks Treatment of micro polluted water | [80] |
| Volcanic Rock | | $NH_4$-N: 94.2% $PO_4$-P: 81.7% | | |
| Ceramsite | | $NH_4$-N: 95.9% $PO_4$-P: 69.6% | | |
| Conventional Filter Media (Synthetic) | | | | |
| **Type of Filter Media** | **Operating Condition** | **Removal Efficiency** | **Additional Remark** | **Reference** |
| Expanded Clay | Temperature: 15 °C Hydraulic Loading: 91 $m^3/m^2$ | TAN nitrification: 100% | Nitrification rate was between 0.1 and 0.2 g TAN/$m^2$ per day Treatment of ordinary tap water | [26] |
| Kaldnes Ring | | TAN nitrification: 80% | | |
| Norton Rings | | TAN nitrification: 60% | | |
| Finturf Artificial Grass | | TAN nitrification: 36% | | |
| Polystyrene | Water velocity: 1.5–6 m/h C/N ratio: 2–9 DO: 0.5–3 $mgO_2$/L Temperature: 15–20 °C | TN: 60.0–70.0% Ammonia: nearly 100% | $NH_4$-N Nitrification: 97.0% $NO_3$-N Denitrification: 71.0% Treatment of municipal wastewater | [81] |

**Table 1.** *Cont.*

| Conventional Filter Media (Synthetic) | | | | |
|---|---|---|---|---|
| **Type of Filter Media** | **Operating Condition** | **Removal Efficiency** | **Additional Remark** | **Reference** |
| GAC-sand dual-media biofilter | Filtration velocity: 8 m/h | $NH_4$-N: 70–74% Steady removal of $NH_4$-N (2.7 mg/L) | Treatment of effluent from the sedimentation tank | [82] |
| GAC (open superstructure) GAC (close superstructure) | EBCT: 18 min Hydraulic Loading Rate: 1.67 m/h Temperature: 25 °C | Near-complete removal of $NH_4$-N in both types of GAC Ammonium: >97.0% | Treatment of raw water | [83] |
| GAC | Filtration velocity: 0.12–0.25 m/h pH: 7.5 | Turbidity: 75.0% DOC: 100% TN: 37% TP: 74% Color: 95% | Treatment of raw stormwater Higher heavy metal removal by GAC compared to anthracite | [55] |
| Polypropylene | Flow rate: 0.3 L/min Airflow rate: 0.3 L/min HRT: 7.5 h | $NH_4$-N: 93.2% Mn: 79.6% | Treatment of contaminated drinking water | [84] |
| GAC | pH: 7.5–8.5 DO: 6.5, 7.5 $mgO_2$/L | DOC: 31.2–34.3% AOC: 51.2–60.6% | Start-up period: 6 months Treatment of pre-treated lake water | [85] |
| Expanded Clay | DO: 4.5 mg/L Temperature: 15–25 °C Flow rate: 1021.6 $m^3$/d | SS: 92.0% $NH_4$-N: 91.0–93.0% | Treatment of municipal wastewater | [86] |
| GAC | Filtration rate: 7–9 m/h HRT: 8 min | Ammonia: 76.5% DON: 54.5% | Start-up period: 38 days Treatment of river water (Pre-treated with coagulation and sedimentation) | [66] |
| Expanded Clay | C/N ratio: 2.9–3.1 DO: 7.0–7.2 mg/L Filtration Velocity: 1.5–3.9 mg/L | $NO_3$-N: 80.0% TN: 50.0% | Treatment of synthetic wastewater | [87] |
| GAC | ECBT: 50 min Influent flow rate: 48 $m^3$/day | Ciprofloxacin: 22.0% Bezafibrate: 25.0% Ofloxacin: 30.0% Azithromycin: 32.0% Sulfamethoxazole: 35.0% | Treatment of secondary effluent | [53] |
| Light Expanded Clay Aggregate | Hydraulic Loading: 5 $dm^3/m^2$ per day HRT: 4 days Temperature: 0, 4, 8, 25 °C C/N ratio: 0.5, 2.5, 5.0 gC/gN | Total Nitrogen: 53.7% Organic Compound: 79.7% | Ammonium nitrogen nitrification efficiency: 50.9% Nitrates and nitrites denitrification efficiency: 99.2% Treatment of wastewater from de-icing airport runway | [88] |
| Polyurethane | C/N ratio: 3–5.6 HRT: 1.8 h, 2.7 h, 3.5 h, 4.2 h DO: 0.3–0.8 mg/L Temperature: 25–28 °C | Total nitrogen: 67–96.5% | Low effluent total nitrogen concentration (0.68 mg/L) Treatment of micro-polluted water | [89] |

**Table 1.** *Cont.*

| Organic Media | | | | |
|---|---|---|---|---|
| **Organic Material** | **Operating Condition** | **Removal Efficiency** | **Additional Remark** | **Reference** |
| Wheat Straw | Temperature: 11 °C Influent rate: 2.7 L/day | TSS: 89.0% Oil & Grease: 76.0% COD: 37.0% $NH_4$-N: 20.0% TKN: 15.0% | Treatment of dairy wastewater | [49] |
| Fibrous Peat | Hydraulic Loading Rate: 180 L/m²d | $BOD_5$: 96.0% $COD_t$: 84.0% TSS: 94.0% | Treatment of domestic strength wastewater | [90] |
| Wood chips | Influent flow rate: 15 mL/min | Nitrate: 99.0% | Treatment of aquaculture wastewater | [30] |
| Wheat Straw | Volumetric loading rate: 340–1380 gN/m³d | | | |
| Mix of peat and wood chips | Aeration rate: 3.4–34 m³/m²/h Filtration Velocity <0.5 m³/m²d | TSS: 98.0% $BOD_5$: 99.0% | Treatment of piggery wastewater | [91] |
| Mixture of endemic tropical woodchips and natural fibers | Hydraulic rate: 0.3 m³/m²d Aeration rate: 0.68 m³ air/m²/h | $BOD_5$: 98.7% COD: 84.0% Faecal Coliform: 99.9% Total Coliform: 99.9% Helminth eggs: 96.4% | Treatment of municipal wastewater | [56] |
| Wild Thorn | Superficial Flow Rate: 15 m³/m²d Temperature: 36–40 °C | $BOD_5$: 76.0% | Treatment of municipal wastewater | [92] |
| Arum Plant | | $BOD_5$: 71.0% | | |
| Date Palm Bark | | $BOD_5$: 62.0% | | |
| Pruning waste of *Caesalpina pulcherrim* and *Jacaranda mimosifolia* | Hydraulic Loading Rate: 0.078 m³/m²/d Aeration Rate: 10 m³/m²/h Temperature: 22 °C | $BOD_5$: 97.0% COD: 71.0% TKN: 93.0% TSS: 95.0% VSS: 96.0% Helminth eggs: 100% Fecal Coliform: 4 log unit | Start-up period: 15 days Treatment of school wastewater | [93] |
| Brewer's Spent Grains | HRT: 100 min Influent nitrate loading: 200 mg/L pH: 7.5–7.9 | NO-3N level was always below the acceptable limit | Start-up period: 1 month Treatment of groundwater | [59] |
| *Agave* waste fibers | Aeration rate: 0.62 m³/m²/h Hydraulic loading rate: 0.27–1.34 m³/m²/d | BOD: 92.0% COD: 79.7% Helminth eggs: 99.9% Fecal coliforms: 99.9% TSS: 91.9% | Start-up period: 3 months Treatment of municipal wastewater | [31] |

**Table 1.** *Cont.*

| Organic Media | | | | |
|---|---|---|---|---|
| **Organic Material** | **Operating Condition** | **Removal Efficiency** | **Additional Remark** | **Reference** |
| Rice straw | Hydraulic rate: 4.8–12 $m^3$/d | $BOD_5$: 81.5%<br>COD: 79.7%<br>TSS: 82.4%<br>TN: 50.2%<br>TP: 41.9% | Start-up period: 2 weeks<br>Treatment of raw sewage | [62] |
| Wood chips of orange tree | | $BOD_5$: 66.7%<br>COD: 64.6%<br>TSS: 68.3%<br>TN: 45.0%<br>TP: 32.5% | | |
| Date palm fiber | | $BOD_5$: 88.3%<br>COD: 88.3%<br>TSS: 86.6%<br>TN: 55.0%<br>TP: 50.5% | | |
| Mesquite wood chips | Hydraulic Loading Rate: 1.07 $m^3/m^2$/d<br>Aeration rate: 0.62 $m^3$ air/$m^2$/h | $BOD_5$: 92.0%<br>COD: 78.0%<br>TSS: 95.0%<br>Fecal Coliform: 4 units | Start-up period: 60 days<br>Treatment of municipal wastewater | [29] |
| Fibrous Carrier and biological ball | Hydraulic Retention Time: 24 h | TN: 37.0–44.0%.<br>COD: 70.4–80.2%<br>P: 22.3–60.5% | Treatment of heavily polluted river water | [94] |
| *Ficus benjamina* wood chips | Hydraulic loading: 0.18–0.37 $m^3/m^2$d. | COD: 91.0%<br>Metformin: 94.0%<br>Ciprofloxacin: 81.0%<br>$NH_3$-N: 81.0% | Start-up period: 80 days<br>Treatment of domestic wastewater | [18] |
| Wood Chips | Hydraulic rates: 0.5–1.5 $m^3/m^2$/d | COD: 80.0%<br>Volatile solids: 40.0–63.0% | Treatment of domestic wastewater | [57] |
| Peanut shells | | | | |
| *Arundo donax* | Air-water ratio: 4:1<br>Nitrate Recycling Ratio: 150%<br>Temperature: 18–25 °C<br>Filtration rates: 1.35 m/d, 3.34 m/d | $NH_4$-N: 99.0%<br>TN: 68.8% | Start-up period: 17 days<br>Treatment of rural domestic sewage | [61] |

Quartz sand and anthracite have been widely utilized as the biofilter media in water and wastewater treatment due to their advantages such as cost-effectiveness and easy availability [32]. Suprihatin et al. [54] found that the stable properties and smaller size of sand promote the effective contact between the pollutants and the biofilm. Aslan and Cakici [71] examined the use of sand as filter media in the biological denitrification of drinking water and obtained an average of 94.0% $NO_3$-N removal efficiency while Yang et al. [74] removed 96.2% of iron, 97.7% of manganese, and 98.2% of arsenic in the treatment of groundwater. A few studies have compared the performance of using sand and anthracite filter media to other alternative media such as GAC and zeolite in the past few years [55,70]. Chang et al. [70] investigated the application of zeolite and sand as filter media in a lab-scale biological aerated filter in the treatment of textile wastewater. It was observed that zeolite media shows better performance in terms of COD, SS, and TKN removal. The superior performance of zeolite is due to the $NH_4$-N ion exchange ability, which favors the growth of autotrophic nitrifiers, thus promoting the TKN removal. Mohammed et al. [55] demonstrated the treatment of raw stormwater by using GAC and

anthracite as the filter media under pH 7.5 and a filtration velocity of 0.12–0.25 m/h. The GAC filters show remarkable removal performance in terms of turbidity, DOC, TN, TP, and color with a higher removal efficiency in terms of organic matter and heavy metal compared to anthracite. The superior performance of GAC compared to other alternative media is largely due to the higher specific surface area, adsorption capacity, porosity, and surface roughness that support a denser microbial population compared to sand and anthracite [20,53].

Other than GAC, sand, and anthracite, materials such as expanded clay, polystyrene, polypropylene, and polyurethane also show good treatment performance as biofiltration media. Additionally, some researchers also investigated the use of multi-media by combining different materials as supporting media for a biofilter. Cheng et al. [75] conducted a pilot-scale experiment by using a combination of manganese, sand, and anthracite as the filter media operating under a temperature of 8 °C and pH 7 for the removal of iron, manganese, and ammonia from groundwater. The removal efficiencies of 90.8% of ammonia, 95.5% of iron, and 95.9% of manganese were achieved in their study. Yu et al. [82] also reported on full-scale biofiltration for drinking water treatment using GAC and sand dual media in achieving a steady removal of $NH_4$-N. In their finding, 57.0% of $NH_4$-N is removed through complete nitrification while 21.5% of $NH_4$-N is partially nitrified to $NO_2$-N.

With organic materials, Ghazy et al. [62] utilized agricultural waste materials such as rice straws, date palm fibers, and wood chips of an orange tree as the biofilter media in the municipal wastewater treatment operated under a hydraulic rate of 4.8–12 m$^3$/d. All of the materials show remarkable removal efficiency ranging from approximately 32.0% to 89.0% in terms of BOD$_5$, COD, TSS, TN, and TP. Vigueras-Cortés et al. [31] also noticed the good performance of *Agave* fibers as a filter media operating under a consistent aeration rate of 0.62 m$^3$/m$^2$/h in treating municipal wastewater with removal efficiencies of 92.0%, 79.7%, 91.9%, 98.0%, and 99.9% for BOD, COD, TSS, helminth eggs, and fecal coliforms, respectively. Moreover, Zhao et al. [61], utilizing *Arundo donax* as the filter media and an external carbon source in the anoxic/oxic biofilter, witnessed that the carbon releasing characteristics of the organic materials tend to exist as an external carbon source in the biofiltration system, thus promoting the denitrification process. The biofilter was operated under a temperature of 18–25 °C, an air water ratio of 4:1, and filtration rates of 1.35 m/d for the oxic column and 3.34 m/d for the anoxic column. From their study, the biofilter successfully removed 99.0% of $NH_4$-N and 68.8% of TN from low C/N rural domestic sewage. Based on our review, biofiltration with organic media shows better performance in terms of the start-up period, denitrification efficiency, and organic matter removal compared to the conventional media biofilter. The start-up period or the biofilm formation on the conventional filter media takes around 20 days to 6 months while organic filter material requires a shorter period, i.e., ranging from 15 days to 3 months. The better start-up performance is believed to be attributed to the characteristics of organic media that provide sufficient surface for the attachment of a microbial community. Some of the organic materials comprise similar properties to GAC, such as large specific surface area, high porosity, and good adsorption ability due to the presence of functional groups, which promote the attachment of a biofilm [18].

Furthermore, the presence of cellulose, hemicellulose, and lignin in some of the organic materials offers a sustainably slow release of carbon sources and structure that supports the growth of denitrifiers and biofilm [95]. Moreover, the denitrification process is usually facilitated by the presence of a sufficient organic matter concentration in the treatment system; however, most of the treated water does not exhibit a sufficient C/N ratio and organic matter for the denitrification process. Therefore, an additional carbon source is needed to maintain the appropriate C/N ratio and organic substrate in the treatment system [59]. According to Zhao et al. [61], the carbon releasing ability of organic materials acts as a carbon source to promote the growth of denitrifiers, which then enhances the denitrification process and total nitrogen removal. This observation is supported by the

study by Chang et al. [94] which reported on the greater relative abundance of denitrifiers found in organic media compared to inorganic media. Other than that, the presence of chemical functional groups in organic materials facilitates chemical binding and improves the adsorption capacity of organic contaminants [57].

### 3.3. Drawbacks of a Conventional Media Compared to an Organic Media Biofiltration System

The conventional filter materials widely utilized by several researchers throughout the past few years include gravel, sand, anthracite, GAC, expanded clay, and plastic media [26,55,69]. Although conventional media biofilters show remarkable performance in various water and wastewater treatments, there are numerous notable drawbacks and limitations worthwhile for consideration in further investigation. Figure 2 illustrates the main drawbacks of conventional media compared to organic media biofiltration systems.

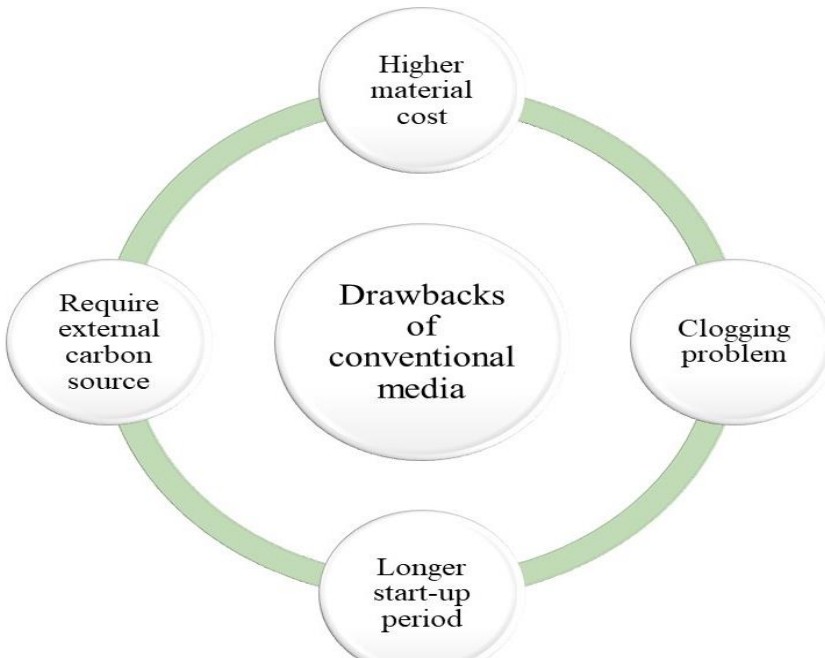

**Figure 2.** Drawbacks of conventional media compared to organic media biofiltration system.

Tejedor et al. [57] acknowledged that the cost of conventional filter materials such as gravel could reach up to 50% of the total investment cost of the biofiltration system, and is 100 times higher compared to organic waste materials. Sharma et al. [68] reported that the costs of conventional filter materials such as ceramic, anthracite, and GAC are 10.7 C\$/m$^3$ ($\approx$8.44 US\$/m$^3$), 12 C\$/m$^3$ ($\approx$9.46 US\$/m$^3$), and 55–60 C\$/m$^3$ ($\approx$43.35–47.29 US\$/m$^3$), respectively, which are much more expensive compared to organic waste materials that are abundantly available as agricultural by-products. For example, coconut fibers as an organic filter media are natural agricultural by-products that are widely available in Southeast Asian countries such as Malaysia [63]. Utilizing these organic waste materials as an alternative to the conventional filter media not only reduces the operational and investment costs but also cuts down the natural solid waste generated throughout the world.

In addition, the accumulation of biomass in conventional filter media creates several downsides such as clogging, large flow resistance, and poor permeability of the biofilm, which result in a decline in the removal performance of biofilter [96]. As an alternative, according to few reports [18,93], utilizing organic materials as the filter media tends to reduce the clogging problem due to the achievement of a balance between degradation of the microorganisms and the retention of solids in the filter bed under lower organic loading. This observation is supported by Chang et al. [94] stating that organic media,

which have higher porosity for high loading rates and sufficient surface for the growth of the biofilm, show better potential to reduce the clogging problems compared to inorganic filter media.

A conventional biofiltration system is further associated with the issue of a longer start-up period during the operation process. The natural biofilm present in the biofilter system comprises several mechanisms, which cause the biofilm to grow at a slower rate and result in a longer start-up period [97]. This limitation is supported by various researchers, as the maturation and acclimatization of biofilm varies, ranging from one month to half a year depending on the operational parameters [23]. Furthermore, biological nitrogen removal in a conventional biofiltration process involves heterotrophic denitrification which requires external carbon sources for electron supply; this includes methanol, acetate, glucose, and ethanol especially in treating sewage with a low pollutant concentration and a low C/N ratio. The accuracy of the additional quantities of external carbon sources remains one of the operational challenges as insufficient organic matter during the treatment process can reduce denitrification efficiencies, leading to the discharge of excessive nitrogenous compounds into the river, while the overdosing of external carbon sources causes an increase in the COD level in the treated effluent [98]. As a solution, organic media are suitable for the solid-phase denitrification process as the main components of plant-based organic materials such as cellulose and hemicellulose, which can be hydrolyzed to form organic acids, acting as a natural carbon source to promote the denitrification process and the growth of denitrifying organisms [61]. Compared to a conventional denitrification biofilter, a solid-phase denitrification biofilter does not involve a costly and complex control system and at the same time minimizes the risks such as under and overdosing of external carbon sources [98].

## 4. Challenges of Using Organic Filter Media

Based on our review, organic materials show promising characteristics as biofilter media in terms of treatment performance; however, not all organic materials are suitable for biofiltration. Zhao et al. [61] commented that plant-based wastes such as peanut shells contain smooth surfaces and smaller specific surface areas, which are relatively difficult for the attachment of biofilm, while organic materials such as rice straws tend to release carbon at a faster rate, causing the prolongation of the denitrification process.

A study conducted by Muliyadi et al. [99] stated that although banana stem media can filter out the excess solid particles from domestic wastewater, the inherent organic properties cause the development of bacteria and the occurrence of natural decay with the increase of exposure time to the water. The degradation rate of the organic media limits the operational lifespan and the sustainability of the organic biofilter. Saliling et al. [30] discovered that the expected lifetimes of wood chips and wheat straws organic media are 1.2 and 0.5 years, respectively, which are much lower compared to the conventional filter media. Extra steps and additional costs are needed in replacing the filter media once they reach their operating lifespan.

Other than that, the degradation of organic materials can cause the leaching of humic acids which increases the concentration of certain contaminants in the treated water such as color, phosphorus, and COD [93]. Bash AlMaliky and Qahtan ElKhayat [92] witnessed the clogging problem and the existence of flies surrounding the organic filters, which are unfavorable in the treatment process. Moreover, there are also relatively few organic filter materials that have been investigated and accessed, which restrict their full-scale applications in many areas and regions such as semi-arid and arid zones [29]. Hence, it can be concluded that the lifespan, decay, and degradation properties of organic materials may also influence the performance of biofiltration. Therefore, future studies should focus on investigating different types of organic filter media that show good and long-term operation performances.

## 5. Research Prospects

Based on this review, many intensive studies are needed to overcome the limitations or drawbacks of biofilters. Unlike conventional filter material that could reach up to 50% of the operational cost, utilizing organic materials or plant-based wastes as the alternative filter media should be focused on in the future due to their advantages such as cost-effectiveness and environmental friendliness. The characterization of organic material in terms of physicochemical and morphological properties and their respective potential as biofilter media is notably important to provide structural stability in the cleaning process. The physicochemical and morphological characterization of established organic media can be a significant reference for researchers when selecting suitable alternative organic filter media. Another limitation of the biofilter is the time-consuming acclimatization of a biofilm on the new filter medium, which results in a longer start-up period; thus, further studies should be focused on identifying a suitable method to reduce the start-up period and enhance the biofilm formation on the filter media.

Further investigation in the future on the optimum operating conditions for simultaneous nitrification and denitrification (SND) in a single biofilter reactor will also be relevant as the operating conditions for the SND process are difficult to obtain and maintain throughout the treatment process. Identification of these suitable environmental conditions that promote the coexistence of nitrifiers and denitrifiers is important to ensure the microbial stratification and feasibility of the SND process. Moreover, there is a lack of studies that apply an organic material as a carbon source in the SND biofiltration system. Therefore, studies should focus more on investigating alternative types of organic materials that are suitable as biodegradable media, which have carbon releasing ability, that ensure the sufficient supply of donor electrons for the denitrifying organisms in the biofiltration system.

Studies also need to be conducted on the cold-water temperature effects on the removal performance of biofilters. Currently, there is a lack of novel methods to overcome the limitation in terms of the decrease in bacteria growth rate due to the cold-water temperatures. Moreover, the assessment of the kinetic analysis of organic media biofilter removal performance is an aspect that is least considered. Extensive analysis is needed on the development and modeling of the kinetics analysis for the treatment process in an organic media biofilter, which serves to clarify the removal mechanism of the organic media biofiltration treatment system. Further characterization of the microbial community present in an organic media biofilm concerning their role in pollutant removal is needed as the microbial species is highly related to the biofilter removal performance. Up to date studies regarding the microbial community present on the organic media and their relationship with the operational conditions of the biofilters are still rare.

## 6. Conclusions

Biofiltration systems show great potential in water and wastewater treatment due to their remarkable performance in treating various types of pollutants in comparison to physical and chemical techniques. However, there are limitations and drawbacks to operating biofilters. Factors such as dissolved oxygen concentration, organic loading rate, hydraulic retention time, temperature, and filter media all affect the dynamics and mechanisms of biological activity, impacting the overall removal performance of biofilters. Based on our review, organic materials show promising potential as an alternative biofilter media due to advantages such as larger surface area, higher porosity, physiochemical stability, non-toxic, higher adsorption ability, lower cost, and environmentally friendly. Furthermore, utilizing organic materials as the supporting media and external carbon source promotes the start-up and denitrification performance. The suitability and the biodegradability of the organic filter media should be considered in biofiltration design as this will affect the lifespan and the resulting performance of the organic biofilter media.

**Author Contributions:** Conceptualization, Z.Z.L., N.S.Z. and E.L.Y.; writing—original draft preparation, Z.Z.L. and N.S.Z.; writing—review and editing, N.S.Z., E.L.Y., A.S., R.B., A.B.H.K. and D.D.P.; supervision, N.S.Z., E.L.Y. and A.S.; project administration, N.S.Z. and A.S.; funding acquisition, N.S.Z. and A.S. All authors have read and agreed to the published version of the manuscript.

**Funding:** This study was funded by the Ministry of Higher Education (MOHE), Malaysia (grant number FRGS/1/2019/TK01/UTM/02/11) and Universiti Teknologi Malaysia (UTM) (grant number Q.J130000.2651.16J76).

**Institutional Review Board Statement:** Not applicable.

**Informed Consent Statement:** Not applicable.

**Data Availability Statement:** Not applicable.

**Acknowledgments:** The authors also thank Universitas Nahdlatul Ulama Surabaya for supporting the current work.

**Conflicts of Interest:** The authors declare that they have no known competing financial interests or personal relationships that could have appeared to influence the work reported in this paper.

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
