# Peer review of "Shifting from Conventional to Organic Filter Media in Wastewater Biofiltration Treatment: A Review"

_applsci, doi:10.3390/app11188650_

Round 1

Reviewer 1 Report

  1. The presented paper is a review, which corresponds to Aims and scope of the journal
  2. The paper looks well organized and has solid structure
  3. It is not clear why "Fibrous Carrier and biological ball" is considered as organic carrier. 
  4. It may recommended to reveal not only the removal efficiency in table 1, but also mean influent and effluent values to make the comparison of various types of media more precise.
  5. The paper may look better if the author showed the research results of their own
  6. Language of the paper needs slight corrections

Author Response

Reviewer #1

The presented paper is a review, which corresponds to Aims and scope of the journal. The paper looks well organized and has solid structure

Our response: The authors thank the reviewer for the positive comments and suggestion.

It is not clear why "Fibrous Carrier and biological ball" is considered as organic carrier.

Our response: The authors thank the reviewer for the suggestion. This study was carried out by the researcher (Chang et al.,2019, 94 according to the references) which compare the treatment performance of inorganic material and organic material as the filter media. In his study, fibrous carrier and biological ball are categorized as organic material.

It may recommended to reveal not only the removal efficiency in table 1, but also mean influent and effluent values to make the comparison of various types of media more precise.

Our response: The authors thank the reviewer for the suggestion. The authors believe the provided removal efficiency are sufficient and can be easily understood from the review summary by the readers. However, the suggestion from the reviewer will be take into consideration for future publications.

The paper may look better if the author showed the research results of their own

Our response: The authors thank the reviewer for the suggestion. However, the research works of the author is currently still in the preliminary stage. The suggestion from the reviewer will be take into consideration for future publications.

Language of the paper needs slight corrections

Our response: The authors thank the reviewer for the suggestion. The authors have carefully checked, revised and improve the overall spelling and grammar of the manuscript.

Reviewer 2 Report

  1. 3 “Studies have also been focused on operating biofilters under different conditions such as filter media, temperatures, backwash regimes, and dissolved oxygen concentrations to achieve remarkable pollutants removal performance.” – add references.
  2. 3” Other than that, there exists also an increasing interest in exploring suitable alternative filter materials such as wood chips, wheat straws, and natural fibers [15]” – add more references.

p.3 - I would suggest to better justify a need for this review as I could see a contradiction in the statement below as activated carbon and anthracite are also made of organic materials. “This is further motivated by the local availability of these proposed organic materials and their similar physicochemical properties compared to the conventional media such as activated carbon and anthracite. Although there are review papers [16–19] discussing some aspects of biofilters, a comprehensive review on the challenges and prospects from the perspective of shifting the use of conventional filter media to that of organic is missing.”

Fig. 1 – I would suggest to use full terms in this figure. Also, would the feed water quality another important factor which affects biofiltration efficiency?

P.4 “Granular activated carbon (GAC), quartz sand, and anthracite are the frequently used conventional filter media in biofilter systems” – reference each material.

P.4 “Tejedor et al [38] found that the organic filter materials should contain favorable characteristics such as large surface area, porous, physiochemically stable (presence of hemicellulose/lignin content), non-toxic, higher adsorption ability, and contain functional groups such as phenolic hydroxyls, carboxylic or methoxyl.” – revise, this sentence is not grammatically correct.

Section 3.2 – I would suggest more profound discussion and comparison of different presented in table 1 with respect to operating conditions, type of biofilter, type of feed water etc.

P.10 – “Quartz sand and anthracite have been widely utilized as the biofilter media in water and wastewater treatment due to their advantages such as cost-effectiveness and easy to handle. S” -  reference this statement.

  1. 10 “ion of drinking water and obtained an average of 94.0% NO3-N removal efficiency while Yang et al [54] removed 96.2% iron, 97.7% manganese, and 98.2% arsenic in the treatment of groundwater.” - should be of iron, of manganese, and so on. This pertains to other similar of the manuscript.

P.10 “In the past few years, several studies had compared the performance of using sand and anthracite to other alternative media such as GAC and zeolite.” – first, reference these studies, Also, revise, as this sentence is not grammatically correct.

I would move section 3.3. “3.3. Drawbacks of Conventional Media Compared to Organic Media Biofiltration System” to the front because it looks like as justification of this review.

p.11 “The conventional filter materials widely utilized by several researchers throughout the past few years include gravel, sand, anthracite, GAC, expanded clay, and plastic media.” – reference this statement.

P.12 “As an alternative, according to many reports [7, 63],” – many means multiple number, but the authors cited only two.

Author Response

Reviewer #2

3 “Studies have also been focused on operating biofilters under different conditions such as filter media, temperatures, backwash regimes, and dissolved oxygen concentrations to achieve remarkable pollutants removal performance.” – add references. 3” Other than that, there exists also an increasing interest in exploring suitable alternative filter materials such as wood chips, wheat straws, and natural fibers [15]” – add more references.

Our response: The authors thank the reviewer for the suggestion. The authors have included some references for these two sentences in page 2. 

p.3 - I would suggest to better justify a need for this review as I could see a contradiction in the statement below as activated carbon and anthracite are also made of organic materials. “This is further motivated by the local availability of these proposed organic materials and their similar physicochemical properties compared to the conventional media such as activated carbon and anthracite. Although there are review papers [16–19] discussing some aspects of biofilters, a comprehensive review on the challenges and prospects from the perspective of shifting the use of conventional filter media to that of organic is missing.”

Our response: The authors thank the reviewer for the comment. The authors have revised the particular statements in page 2.

Fig. 1 – I would suggest to use full terms in this figure. Also, would the feed water quality another important factor which affects biofiltration efficiency?

Our response: The authors thank the reviewer for the suggestion. The authors have revised the words in the figures into full terms in page 3. For the feed water quality, although it will affect the biofiltration efficiency, however, it exists as a controlling factor (type of water treated) and it is also related to the organic loading rate in most of the studies reported. Therefore, the authors did not include it as one of the important or impactful factors.

P.4 “Granular activated carbon (GAC), quartz sand, and anthracite are the frequently used conventional filter media in biofilter systems” – reference each material.

Our response: The authors thank the reviewer for the suggestion. The authors have included references for each material in page 4.

P.4 “Tejedor et al [38] found that the organic filter materials should contain favorable characteristics such as large surface area, porous, physiochemically stable (presence of hemicellulose/lignin content), non-toxic, higher adsorption ability, and contain functional groups such as phenolic hydroxyls, carboxylic or methoxyl.” – revise, this sentence is not grammatically correct.

Our response: The authors thank the reviewer for the suggestion. The authors have revised this sentence in the manuscript in page 4.

Section 3.2 – I would suggest more profound discussion and comparison of different presented in table 1 with respect to operating conditions, type of biofilter, type of feed water etc.

Our response: The authors thank the reviewer for the suggestion. The authors have improved and included the operating conditions, types of biofilter and types of treated water in the discussion and comparison of different presented in table 1 in page 10 and 11.

P.10 – “Quartz sand and anthracite have been widely utilized as the biofilter media in water and wastewater treatment due to their advantages such as cost-effectiveness and easy to handle. S” -  reference this statement.

Our response: The authors thank the reviewer for the suggestion. The authors have included that references in page 10.

10 “ion of drinking water and obtained an average of 94.0% NO3-N removal efficiency while Yang et al [54] removed 96.2% iron, 97.7% manganese, and 98.2% arsenic in the treatment of groundwater.” - should be of iron, of manganese, and so on. This pertains to other similar of the manuscript.

Our response: The authors thank the reviewer for the suggestion. The authors have revised on these throughout the manuscript and page 10.

P.10 “In the past few years, several studies had compared the performance of using sand and anthracite to other alternative media such as GAC and zeolite.” – first, reference these studies, Also, revise, as this sentence is not grammatically correct.

Our response: The authors thank the reviewer for the suggestion. The authors have included that references and revised the sentences in page 10.

I would move section 3.3. “3.3. Drawbacks of Conventional Media Compared to Organic Media Biofiltration System” to the front because it looks like as justification of this review.

Our response: The authors thank the reviewer for the suggestion. However, the authors still decided to remain this under section 3.3 as it will serve as a summary and justifiy why there is a need to shift to organic materials after comparing their performance on section 3.2.

p.11 “The conventional filter materials widely utilized by several researchers throughout the past few years include gravel, sand, anthracite, GAC, expanded clay, and plastic media.” – reference this statement.

Our response: The authors thank the reviewer for the suggestion. The authors have included the references in page 11.

P.12 “As an alternative, according to many reports [7, 63],” – many means multiple number, but the authors cited only two

Our response: The authors thank the reviewer for the suggestion. The authors have revised and removed the words “many reports” in page 12.

Round 2

Reviewer 2 Report

I am satisfied with authors' responses to my comments.